# Exploring Bayesian Deep Learning Uncertainty Measures for Segmentation of New Lesions in Longitudinal MRIs

## Abstract

In this paper, we develop a modified U-Net architecture to accurately segment new and enlarging lesions in longitudinal MRI, based on multi-modal MRI inputs, as well as subtraction images between timepoints, in the context of large-scale clinical trial data for patients with Multiple Sclerosis (MS). We explore whether MC-Dropout measures of uncertainty lead to confident assertions when the network output is correct, and are uncertain when incorrect, thereby permitting their integration into clinical workflows and downstream inference tasks.

**Keywords:** Multiple Sclerosis, New and enlarging lesions, longitudinal MRI, Bayesian Deep Learning.

## 1. Introduction

The presence of new or enlarging (NE) T2-weighted lesions in longitudinal Magnetic Resonance Images (MRIs) of patients with Multiple Sclerosis (MS) is an important marker of new disease activity, and therefore use for assessing patient worsening and treatment effects. Challenges in the automatic segmentation of NE lesions from longitudinal MRIs are abundant as NE lesions tend to be small, and real anatomical change must be differentiated from artifactual changes (e.g. due to registration errors and brain atrophy) in subtraction images between MRIs acquired at different timepoints (Molyneux et al., 1999) (see Figure 1). Given the difficulty of the task, a deterministic output should be interpreted with care, particularly as errors in segmentation can have significant clinical impact. For example, the false segmentation of a **single** NE lesion when none are present (or vice versa) results in a false conclusion regarding new disease activity and treatment effect. Recent Bayesian Deep Learning (BDL) methods such as MC-Dropout (Gal and Ghahramani, 2016) and Prob-U-Net (Kohl et al., 2018) allow the model to estimate the uncertainties associated with network output. This can lead to better adoption of these models into a clinical flow.

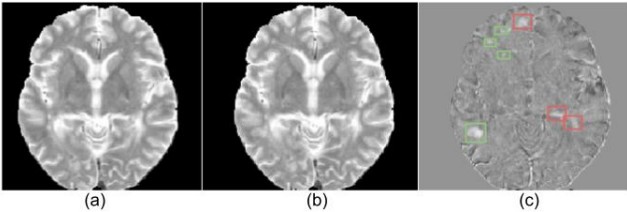

Figure 1: Patient MRI (a) at reference and (b) at follow-up timepoints. The resulting subtraction image (c). Green boxes represent ground truth NE lesions. Red boxes reflect other bright spots resulting from competing differences.

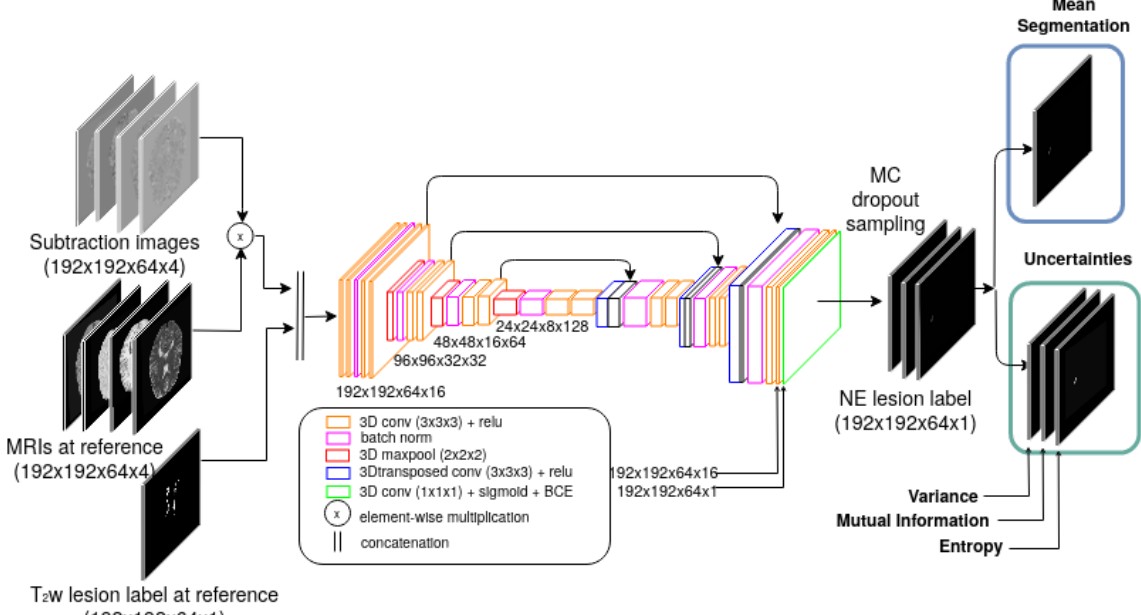

Figure 2: Architecture of the segmentation network. All operations including convolution, max-pooling, and up-sampling and dropout are applied to 3D volumes.

These BDL models have been examined for different medical imaging tasks like MS T2 lesion segmentation (Nair et al., 2020), chest radiograph assessment (Ghesu et al., 2019), MS progression prediction (Tousignant et al., 2019), etc. These papers report an improved performance when evaluated on its most certain predictions. In this paper, we verify these claims for the challenging task of NE lesion segmentation using a modified 3D U-Net (Çiçek et al., 2016) architecture and using MC-Dropout at test time to generate uncertainties associated with its output (e.g. entropy, sample variance and mutual information) (Gal et al., 2017). Our experiments on a large, proprietary, multi-center, multi-modal, clinical trial dataset consisting of 1677 multi-modal scans concur the finding in the previous studies and shows that model performance improves when evaluated on its most certain predictions.

## 2. Method

Given its popularity, a 3D U-Net (Çiçek et al., 2016) trained with dropout is designed to perform voxel-level segmentation of NE lesions from MRI sequences acquired a year apart (Figure 2). During testing, the same input is passed the network N times to obtain N MC-samples and estimate the uncertainty using various uncertainty measures: entropy, mutual information and sample variance (Gal et al., 2017; Nair et al., 2020). The mean of the sample segmentation is used to obtain a single prediction.

### 2.1. Filtering based on Uncertainty thresholding

We follow the same procedure followed by previous work (Nair et al., 2020; Ghesu et al., 2019) of evaluating performance of the network output by filtering out its most uncertain prediction. We normalize output uncertainty between 0-1 across the whole dataset. After

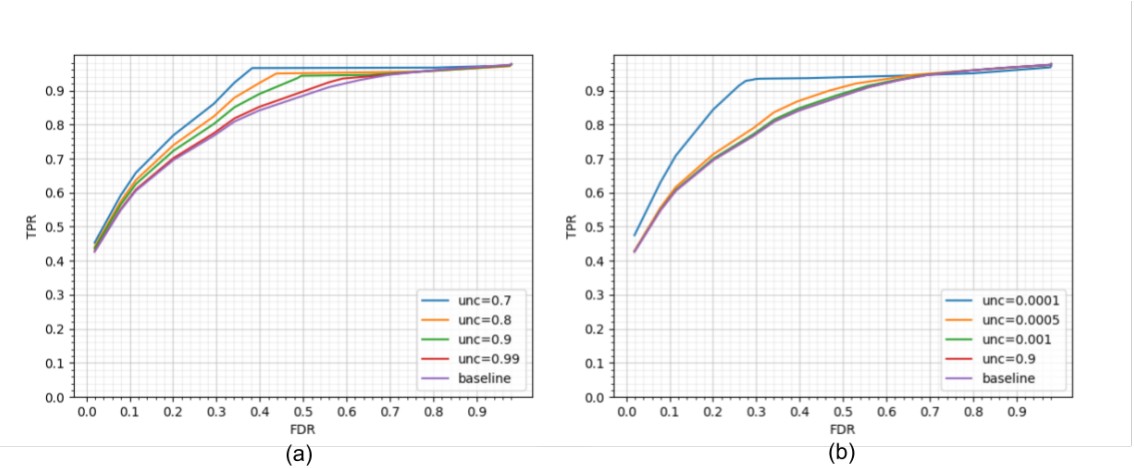

Figure 3: Effect of Uncertainty based filtering on voxel level segmentation ROC for (a) Entropy and (b) Variance measures.

this, voxels are marked as uncertain if they have uncertainty above a threshold value 't', and performance metric (here, ROC curve of TPR vs FPR) is evaluated on only certain voxels. This procedure is repeated for different uncertainty thresholds 't'. AUC of ROC should increase by decreasing uncertainty threshold 't' if incorrect voxels (FPs and FNs) are more uncertain compared to correct ones (TPs).

## 3. Experiments and Results

For this work a proprietary dataset consisting of 1677 images acquired during a large, multi-center, multi-scanner clinical trial was used. The trial was two years long, and multi-modal MR scans were obtained at the end of each year. At each timepoint, T1-weighted (T1w), T2-weighted (T2w), Proton Density-weighted (PDw) and Fluid-attenuated inversion (FLAIR) sequences were available for each patient. Semi-manual expert T2w and NE lesion labels were also included in this dataset. The dataset is randomly divided into a training (60%), validation (20%) a test set (20%).

In Figure 3. effect of uncertainty thresholding for entropy and variance measures is depicted for voxel level segmentation [1]. From these curves, we can see that by decreasing the uncertainty threshold (marking more voxels as uncertain), we can attain higher ROC curve on the remaining voxels.

## 4. Conclusion

In this paper, we verified previous claims (Nair et al., 2020; Ghesu et al., 2019; Tousignant et al., 2019) that deep models tends to give higher performance when evaluated on its most certain output, by experimenting on a more challenging NE lesion segmentation task on large, proprietary, multi-center, multi-model, clinical trial dataset.

---

1. Due to space constraints, we didn't show results for Mutual Information. If the paper is accepted, results for mutual information measures as well as lesion level analysis for all types of uncertainty measures will be presented

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
