# OpenReview forum: "Exploring Bayesian Deep Learning Uncertainty Measures for Segmentation of New Lesions in Longitudinal MRIs"
_MIDL.io/2020/Conference — Submitted to MIDL 2020_

### Official Review · AnonReviewer3 · 2020-02-21
**MCDO for the segmentation of 'New/Enlarging' lesions, well-written, promising results, but proprietary data**

**Rating:** 3
**Confidence:** 4

**Review:**

The paper uses MCDO for uncertainty estimation in the segmentation of 'New/Enlarging' brain lesions. it is well written and the problem statement is clear. With the use of uncertainty information, they leap from deterministic segmentation, which can be perilous in the given medical context, to a probabilistic approach. The validation is valid and the results are promising. However, in an extended version of the paper, I would love to see a more comprehensive list of methods for uncertainty estimation. MCDO is not the only one and it has its failure modes. From a decent comparison of such methods, we can learn more about the nature of this important medical problem as well as the performance of other methods on this real-life application.

One thing I can brag about is the use of a proprietary dataset. It sounds like a good collection but closed-sourceness of medical data gives me a bad taste, always. In my opinion, proprietary datasets, file formats, etc... hinder the progress overall.

In summary, the paper is of interest to the MIDL audience and could benefit from further discussions.

---

### Official Review · AnonReviewer1 · 2020-03-10
**Validated Application for lesion segmentation using Baysesion deep learning**

**Rating:** 2
**Confidence:** 4

**Review:**

Summary: This paper explored the uncertainty measuring in lesion segmentation task. U-Net architecture is used as backbone network. Monte-Carlo Dropout approach is used to measure the uncertainty.  New or enlarging lesion is the main segmentation target, thus subtraction images are also feed to network.

Props: The authors accomplished a complete NE lesion segmentation task using 3D U-Net architecture, and incorporated MC-dropout approaches to measure the uncertainty. Whole paper is well written, easy to follow.

Cons:
1) This work is more like a reproducibility of (a part of) previous work (Nair, MIA, 2020). The major difference is validation dataset. Personally, I think this paper lacks novelty. The authors emphasized the segmentation task for NE lesion is challenging, but they did not give any support for this claim.
2) Since the authors claimed "we develop a modified U-Net", the modified part should be well explained. I cannot find any major difference with the original U-Net architecture except for the input data.
3) More details should be well explained in limited pages. Such as network architecture, detailed filtering.
3) Typo and grammatical errors need to be fixed. Such as 'a test set', 'Figure 3.', "‘t’", 'follow the same procedure followed by...'

Comments:
I think the author can go deeper in uncertainty measuring tasks, not just changing the dataset. As claimed in the abstract, "... thereby permitting their integration into clinical workflows and downstream inference tasks“. More deeper in downstream clinical workflows would be interesting than re-validation of previous works.

---

### Official Review · AnonReviewer4 · 2020-03-10
**Well-motivated short paper with several shortcomings**

**Rating:** 2
**Confidence:** 4

**Review:**

Quality and clarity:
The short paper is well-written and easy to follow.

Significance:
The presented work is very similar to the work of Nair et al., 2020. The difference is mainly the modified segmentation task. Due to the similarity of the problem statement, the methods used, and the significance of the reported result (and the fact that this work is not introduced as a short paper of an existing publication), the benefit for the readership is limited.

Pros:
- The work is well-motivated, and the short paper nicely introduces the problem.
- By focusing on the assertion confidences, this work addresses a critical issue with regards to the clinical integration of DL approaches.

Cons:
- The management of the available space is poor. Space limitations are mentioned as a reason not to show additional results. At the same time, a large figure of a 3D U-Net architecture is presented. The benefit of showing architecture details is minimal, especially because the essential information about the dropout locations is missing. I would prefer seeing additional results than the network architecture.
- The benefit of the proposed approach is unclear. The work mainly shows that ignoring uncertain voxels leads to improved results. As I understand, to observe such a benefit it only requires that some FP/FN voxels express uncertainty, which should be the case for any uncertainty estimation method. The method should thus, at least, be compared to the standard softmax probability (or entropy) output of the network to assess the benefit of the proposed method.
-In the abstract, the work claims: “We explore whether MC-Dropout measures of uncertainty lead to confident assertions when the network output is correct, and are uncertain when incorrect [...]”. I do not see how the results support this claim since the evaluation only requires that the certain voxels are correct (as correctly mentioned in the conclusion).

Minor:
- In the text, the ROC is defined as TPR vs. FPR, whereas in the plot it is TPR vs. FDR.
- Baseline (in Figure 3) is not explained. I assume it to be the absence of an uncertainty threshold.

---

### Official Review · AnonReviewer2 · 2020-03-11
**The paper is well written and clearly motivated. The introduction excites the reader, but as the methodology commences, the paper lacks depth and justifications for different choices in the experimental setup.**

**Rating:** 2
**Confidence:** 5

**Review:**

Pros:
- Well written
- Clearly motivated
- Quality and clarity of the presentation are great
- Experiments are conducted on a very large dataset

Cons:
- Unclarities in methodology (1): for n volumes, you end up with n-1 subtraction images. How can you multiply these elementwise with n volumes?
- Unclarities in methodology (2): How is MC dropout applied, or where is dropout placed in the network?
- Originality: The way the focus was put in this work forces me to question the novelty. I'd have loved to see more details and justifications on the design choices of the network input
- Data (Minor): Why was T2 used instead of FLAIR?
- How would one determine a threshold on uncertainty, which commonly is not between 0 and 1, on a test set for which training set uncertainties is not known?
- What precisely is meant by "at reference"?
- What exactly is the output of the 3D Unet? One segmentation volume, or 3 of them?
- The reported metric: ROC and AUROC are not suitable for my point of view in this context. I'd assume that lesion and background pixels are heavily imbalanced, which calls for the Precision-Recall-Curve and the respective area under it.

---

### Meta-Review · Area_Chair1 · 2020-03-26
**MetaReview of Paper298 by AreaChair1**

**Rating:** 2

**Metareview:**

While the reviewers agree that the paper is well written and that the application is relevant, they also share concerns on the novelty and presentation of this work.

**Paper Type:**

both

---

### Decision · Program_Chairs · 2020-04-11

Reject